# Neurocognitive Functioning and Suicidal Behavior in Violent Offenders with Schizophrenia Spectrum Disorders

**DOI:** 10.3390/diagnostics10121091

**Published:** 2020-12-15

**Authors:** Miriam Sánchez-Sansegundo, Irene Portilla-Tamarit, María Rubio-Aparicio, Natalia Albaladejo-Blazquez, Nicolás Ruiz-Robledillo, Rosario Ferrer-Cascales, Ana Zaragoza-Martí

**Affiliations:** 1Department of Health Psychology, Faculty of Health Science, University of Alicante, 03690 Alicante, Spain; miriam.sanchez@ua.es (M.S.-S.); irene.portilla@ua.es (I.P.-T.); maria.rubio@ua.es (M.R.-A.); natalia.albaladejo@ua.es (N.A.-B.); nicolas.ruiz@ua.es (N.R.-R.); 2Department of Nursing, Faculty of Health Science, University of Alicante, 03690 Alicante, Spain; ana.zaragoza@ua.es

**Keywords:** suicide, neurocognition, schizophrenia, offenders, neuropsychology

## Abstract

Suicide is one of the main premature causes of death in patients with schizophrenia. However, little is known about the relationship between neurocognitive functioning and suicidality in violent offenders with schizophrenia who have been sentenced to psychiatric treatment after committing violent crimes. We examined the neurocognitive functioning of a sample of 61 violent offenders, most of them murderers with schizophrenia who were classified as suicide attempters (*n* = 26) and non-attempters (*n* = 35). We compared the neurocognitive functioning of both groups using a neuropsychological battery. Suicide attempters showed similar performance to non-attempters in a neuropsychological test across all domains of cognitive functioning, memory, attention, verbal fluency, and executive functioning. However, after controlling for demographic and clinical variables, suicide attempters performed better than non-attempters in two planning-related tasks: the Tower of London (*p* < 0.01) and the Zoo Map (*p* < 0.01). Suicide attempters were also characterized as having more family histories of suicidality and as displaying more depressive symptoms and negative symptoms of psychopathology on the Positive and Negative Syndrome Scale (PANSS) scale. These results suggest that suicide attempters have a greater ability to formulate plans and initiate goals directed at making a suicide attempt.

## 1. Introduction

Suicide is a major cause of death among people with schizophrenia and related disorders. It has been estimated that around 5% of these patients die by suicide and between 20 to 50% make at least one suicide attempt during the course of their illness [1,2]. The risk for suicide among people with schizophrenia and related disorders is much more elevated among patients sentenced to compulsory psychiatric treatment and among those exposed to the criminal justice system due to violent crimes [3]. It has been estimated that 6% of murderers suffer from schizophrenia [4,5], and that the risk of committing an act of suicide or homicide is significantly elevated among these patients compared to the general population [6]. Some common risk factors, such as clinical symptoms of illness, hostility, impulsivity, and lack of compliance with medication, may be related to suicide and violent behaviors and exacerbate these tendencies whose course may run in parallel [7].

Suicidal behavior in these populations has been considered as a continuum of the increasing seriousness and lethality of behaviors, moving from thoughts, plans, or wishes to self-injuries and fatal outcomes [8]. Some specific risk factors, such as stresses of imprisonment, mental illness, social deprivation, a history of suicide attempts, and childhood physical abuse, have been demonstrated to play an important role in suicide and self-harm among people in custody, particularly among psychiatric forensic patients with schizophrenia [9,10]. However, most of these relevant environmental correlates of suicide and violence are present in a large number of patients with schizophrenia who will not commit suicide [11].

Studies examining neurobiological correlates of suicidality in criminal samples of patients with schizophrenia-related disorders have shown that certain structural neural abnormalities, including the fronto-temporo-limbic circuits and amygdala, are associated with suicide and self-harm in violent offenders with schizophrenia spectrum disorders [12,13,14]. These regions play an important role in executive functions, inhibition responses, and emotional processing to external stimuli [15,16,17]. Some authors suggest that an inefficient integration of the information during emotion processing may precipitate an aggression in patients with schizophrenia and related disorders. In particular, it has been reported that patients who display information processing deficits show an increased connectivity between the dorsal anterior cingulate cortex and bilateral prefrontal cortex, as well as a decreased functional connectivity between the frontal regions and the putamen and hippocampus regions. These brain abnormalities can be implicated in the failure to regulate negative emotions, such as fear and anger [18]. Moreover, grey matter reductions in the anterior cingulate have been reported as neuroanatomical markers of genetic liability to psychosis, while reductions in the superior temporal gyrus and cerebellum may be interpreted as markers of a first onset of the illness [19].

Neuropsychological studies examining cognitive functioning in violent offenders with schizophrenia spectrum disorders have partially supported neurobiological findings, but it is unclear whether self-harm can be characterized through neuropsychological tests [20]. Some, but not all, studies have reported that higher executive functioning on measures of attention, verbal fluency, and cognitive flexibility [21,22] may influence the ability to initiate and plan suicidal behavior; however, other studies have failed to find such an association [23,24]. The inconsistent results may reflect methodological differences between studies, including sample heterogeneity, study settings, and differences in the neuropsychological test batteries utilized. In addition, most studies to date focused on clinical samples of patients with schizophrenia, with very few studies examining the rates of suicide among patients with exposure to the criminal justice system.

Therefore, the aim of this study was to examine the relationship between neurocognitive functioning and history of suicidality in violent offenders with schizophrenia. We aimed to test the hypothesis that better cognitive functioning of the executive functions will be associated with a history of suicide attempts. To test this hypothesis, we examined a sample of 61 extreme violent offenders with schizophrenia and related disorders sentenced to compulsory psychiatric treatment in a maximum security forensic psychiatric hospital in Spain.

## 2. Materials and Methods

### 2.1. Participants

The present study included a sample of male patients with schizophrenia and related disorders recruited from the Forensic Psychiatric Hospital of Alicante (Spain). This institution is the main psychiatric hospital in Spain, covering 80% of the Spanish population sentenced to compulsory psychiatric treatment by the criminal justice system. The institution has 375 beds for violent offenders with major mental disorders, personality disorders, and mental disability. It provides medium and maximum security facilities for all violent offenders admitted due to court orders or transferred from prisons because of a mental disorder. At the time of the study, there were 275 patients at the institution. Around 34% of patients (*n* = 96) were admitted under psychiatric orders after committing an offense of murder or homicide [25].

Inclusion criteria in the study were that all subjects be male, right-handed, with a primary diagnosis of severe mental illness, including schizophrenia, schizoaffective disorder, or delusional disorder according to the Diagnostic and Statistical Manual of Mental Disorders [26], a prior history of criminal offence leading to compulsory psychiatric admission, Spanish as the mother tongue, and being free of current substance abuse (tested by urine test). Participants with a history of neurological illness (e.g., stroke, Parkinson’s disease), head injury (causing loss of consciousness for more than 30 min), mental retardation defined by using DSM-IV-TR criteria, substance-induced psychotic disorders, and a current presence of severe symptoms of psychopathology as defined by a score ≥ 3 on the Positive and Negative Syndrome Scale (PANSS) at the moment of assessment [27] were excluded. From the 88 patients approached, *n* = 8 (9%) were excluded because they were too ill, or due to active psychopathology at the time of the study. Eleven patients (12.5%) declined participation, *n* = 7 (8%) patients were excluded due to a history of neurological illness or head injury, and *n* = 9 (10.2%) patients were cognitively impaired. The final sample included 61 patients with schizophrenia spectrum disorders.

### 2.2. Clinical Assessment

An experienced psychiatrist at the institution conducted all clinical diagnoses. A psychologist trained in the Structural Clinical Interview for DSM-IV Axis I disorders (SCID-I) [28] also checked the clinical criteria of diagnosis made by a psychiatrist. Age at illness onset was defined as the age during first SCID-verified psychotic symptoms. Psychopathological symptoms were measured using the Positive and Negative Syndrome Scale (PANSS) [29]. Most participants had a long previous history of psychiatric treatment, with at least two or more prior contacts with mental health services (*n* = 51, 83.1%). Using the Global Assessment of Functioning (GAF) [30], the majority of patients were moderately ill (*n* = 35, 68.6%) at the time of the study, with a mean score of 52.14 (SD = 13.92). Over 90% of the total sample were taking antipsychotic medications (14.0% typical, 51.3% atypical, and 29.3% both typical and atypical). Approximately a third (29.3%) were treated with clozapine. Medication information was converted into chlorpromazine dosage equivalents where applicable.

The number of suicide attempts was coded based on the SCID interview and additional information from hospital records. Patients were classified as suicide attempters if they had a prior history of one or more suicide attempts. Patients who had never attempted suicide were classified as non-attempters. Both groups were matched according to age, years of education, primary diagnosis, and violent offenses following incarceration.

### 2.3. Ethical Approval

The study was approved by the Ethics Committee of the Alicante Forensic Psychiatric Hospital (code number HPPA-2885/431-2014), and was conducted according to the 1964 Helsinki Declaration of the Ethical Principles for Medical Research Involving Human Subjects. All participants were mentally capable and legally competent to give written informed consent according to the Spanish Civil Law Procedure (art. 293). Patients were informed that their answers would have no negative consequences and would not affect their privileges, restrictions, or treatment.

### 2.4. Neuropsychological Assessment

Neuropsychological assessment was conducted by a psychologist trained in standardized neuropsychological testing. The battery was designed to measure a broad spectrum of neurocognitive domains shown to be impaired in schizophrenia spectrum disorders. Premorbid IQ was estimated using the National Adult Reading Test (NART) [31]. Current IQ was assessed using the Brief Intelligence Test (K-BIT) [32], which measures cognitive functions through two subtests, namely verbal (vocabulary) and nonverbal (matrix). Attentional control was assessed using the d2 [33]. Episodic, verbal, and working memory were assessed using the subsets of the Wechsler Memory Scale, third edition (WMS-III), and the subscale of Letters and Numbers (WAIS-III). Executive functioning was assessed using the Wisconsin Card Sorting Task (WCST) [34], a frequently used test to measure perseverance, abstract thinking, and set shifting, as well as the computerized Tower of London test (TOL) [35], the Zoo Map subset from the Behavioral Assessment of Dysexecutive Syndrome (BADS) [36], the Trail Making Test [37], and the Stroop Color Word task [38]. Planning abilities were assessed using the TOL, a widely used problem-solving task to assess planning and organization. We used the time to start the first move (latency or planning time) and the number of moves exceeding the necessary number of moves from 10 trials. We also used the Zoo Map, which is considered to be an ecologically valid test of planning and problem-solving ability. Response inhibition and interference resolution were measured with the Stroop task [39]. Finally, verbal fluency was measured using the controlled oral word fluency task (FAS) [40].

## 3. Results

### 3.1. Demographic, Clinical, and Criminal Variables

The demographic, clinical, and criminal variables of patients with (*n* = 35) and without (*n* = 26) lifetime suicide attempts are presented in Table 1. There were no significant group differences in age, age of illness onset, years of education, clinical diagnosis of major mental disorders, and comorbid personality disorders. Most patients had a comorbid antisocial personality disorder. There were no significant differences between the two groups in the rate of prescription of typical versus atypical neuroleptic medications. There were no significant differences in the history of criminal offences. The most common offenses leading to compulsory treatment were murder and homicide (74.7%), followed by attempted homicide (14.8%) and other severe violent offenses, including injuries (14.8%) and sexual assault (4.3%). Suicide attempters had significantly higher numbers of family members who died by suicide (χ^2^ = 4.76, *p* = 0.029). Although not reported in the table, the average length of stay at the institution at the start of the study was 143 months (SD = 81.12, range 6–360 months). The majority of patients had committed their offenses against family members or known victims (71.2%), and 13.4% had been physically or sexually abused as children.

### 3.2. Psychopathology and Psychopathy Clinical Symptoms

Psychopathology and psychopathy symptoms in patients with (*n* = 35) and without (*n* = 26) lifetime suicide attempts are presented in Table 2. There were no significant group differences in the total score, positive symptoms, or complex index of psychopathology according to the PANNS scale. However, suicide attempters had significantly higher PANSS negative symptom scores than non-attempters (*t* = −2.13; *p* = 0.037). Based on these results, the negative symptoms of PANSS were controlled in the regression model. There were no significant differences in psychopathy scores between suicide attempters and non-attempters. Both groups were similar in personality traits of psychopathy (factor 1) and behavioral traits of psychopathy (factor 2). 

### 3.3. Neuropsychological Differences between Offenders with and without Lifetime Suicide Attempts

Results regarding neuropsychological differences between suicide attempters and non-attempters are reported in Table 3. In general, there were no significant differences between the two groups in neurocognitive functioning. Both groups displayed similar cognitive functioning in executive functioning. Suicide attempters and non-attempters showed similar functioning in measures of pre-morbid and current intelligence, memory, verbal fluency, attention, cognitive flexibility, set shifting, and inhibitory control. However, suicide attempters performed better than suicide non-attempters in visuospatial planning and problem-solving tasks (Zoo Map and Tower of London, respectively) in terms of accuracy and time taken. Additionally, suicide attempters showed more clinical symptoms of current depression (*t* = −3.39, *p* < 0.001). No significant differences were found in measures of anxiety.

### 3.4. Predictors of Suicidality

We applied a multivariable binary logistic regression model using prior attempts of suicide (yes or no) as a dependent variable. As predictors of suicidality, we used variables that yielded a statistically significant result on the univariate analyses (i.e., family suicidality, PANNS negative symptoms, Hamilton total depression, Zoo Map Part 1, Zoo Map Part 2, Tower of London extra moves, and Tower of London time in seconds). The forward conditional method was applied using *p*-values < 0.05 as the criteria for model inclusion variables. As can be seen in Table 4, the Zoo Map part 1 (*p* = 0.002), the Tower of London extra moves (*p* = 0.004), and the Tower of London time in seconds (*p* = 0.027) were statistically significant predictors for lifetime suicide attempts between suicide non-attempters and attempters. These variables could explain 76.1% of variance (Nagelkerke R^2^ = 0.761). Finally, we conducted a linear regression model using the number of suicide attempts as the dependent variable. The model yielded the statistical significance (*F* (7,53) = 7.30; *p* = 0.000), with a 42.4% of variance accounted for (R^2^ = 0.424). Again, the Zoo Map part 1 (*p* = 0.011), the Tower of London extra moves (*p* = 0.012), and the Tower of London time in seconds (*p* = 0.015) were significant predictors for lifetime suicide attempts. 

## 4. Discussion

This study examined the relationship between neurocognitive functioning and suicidality in a sample of violent offenders with schizophrenia. Specifically, we investigated whether the nature and extent of suicide attempts were influenced by neuropsychological function. We found that most measures of neurocognition could not predict suicide attempts. We found no significant differences between suicide attempters and non-attempters in measures of intelligence, memory, attention, or executive function. Only planning ability could differentiate between those who had previously attempted an act of suicide and those who had never contemplated suicide attempts. 

Our results are contradictory to the extended literature that suggests that suicide non-attempters display a more preserved cognitive function than suicide attempters [41], and are consistent with previous studies of suicide attempters reporting no relationships between most common neuropsychological test results and suicidality [22,23,24,42].

Research to date has extensively examined the neuropsychological performance of patients with schizophrenia, but very few studies have included planning ability as a measure of assessment, or have employed only a single measure of planning ability [43]. To our knowledge, this is the first study to examine planning ability in relation to suicidality in violent offenders with schizophrenia and exposure to the criminal justice system. Prior studies have demonstrated that exposure to the criminal justice system contributes to an elevated risk of suicide and self-harm, especially among people sentenced to psychiatric treatment and among those with a history of violent offense charges [3]. The risk of suicide is most significant among offenders who have committed criminal offenses against family members [3,44].

In the current study, we found that the negative dimension of psychopathology, a family history of suicidality, depression, and planning were able to differentiate between suicide attempters and non-attempters. However, when we examined the predictors of suicidality in the regression model, only the planning ability emerged as a significant predictor of suicide attempts. Some previous studies conducted in clinical samples of psychiatric patients reported that planning might be a core component of the risk for suicide. For example, Holt et al., in a study of patients with schizophrenia and patients with unipolar major depression who completed a neurocognitive battery focused on executive functions, including planning, found that both groups were impaired in measures of attention, working memory, and planning compared to healthy individuals, but only planning ability differentiated between the patient groups [43]. Similarly, Moniz et al., examining a sample of depressed suicide attempters, reported that planning might contribute to the risk of suicide because it requires intentionality and the organization of each step to effectively commit an act of suicide [45]. Moreover, the preservation of the executive function of planning in suicide attempters might predispose these individuals toward a better ability to formulate plans and initiate goal-directed behavior to commit suicide [22]. The planning ability might be particularly significant within controlled environments, such as a maximum security psychiatric hospital, given that these settings are characterized by restrictive access to methods of suicide [46,47]. In these settings, patients need to develop alternative strategies to initiate goal-directed behavior to commit suicide. Some recent findings suggest that, among people with schizophrenia and related disorders, suicide can be influenced by external and environmental factors, such as the stigma of illness, social rejection, segregation, and intense feelings of regret and guilt, particularly if offenses were committed against family members [3,44]. In addition, demoralization, a syndrome of existential distress occurring in patients with severe conditions that threaten life or integrity of being, such as physical illness or mental disorders, can play an important role in self-harming behaviors and suicide [48]. Demoralization has been associated with feelings of impotence, isolation, and despair [49]. Research has suggested how demoralization is often present in patients with schizophrenia and is a predictor of suicidality, particularly among patients who experience loss of meaning in life and loss of hope, which are common characteristics among violent offenders with schizophrenia spectrum disorders recruited in forensic psychiatric institutions. 

The current study had a number of potential limitations that need to be considered in future studies. These include the modest sample size and the controlled environment of the study. It should be noted that our findings are specific to violent offenders with schizophrenia spectrum disorders, and might not be generalizable to other populations. Despite these limitations, the current study has a number of strengths that increase our confidence in the results, including the homogeneity of the sample, which was restricted to criminal offenders with schizophrenia and related disorders, and the use of multiple datasets from the hospital record system and the clinical history of each patient to collect the past history of suicide attempts. Additionally, we used a large and comprehensive neuropsychological battery covering all related domains of neuropsychological function in schizophrenia. In addition, we used several planning measures, and thus, our results cannot be considered as a spuria estimation. The current findings are promising for clinical practice by providing new opportunities for suicide prevention among patients processed by the criminal justice system. Our results suggest that is important to identify patients with a prior history of suicidality, depressive symptoms, and a high ability of planning. These factors might contribute to the improvement of the actual risk assessment methods of suicide prediction. The inclusion of new markers of suicide, such as neuropsychological tests, might also offer new clinical markers of suicide intentionality. However, whether these measures can predict suicide incidents in forensic psychiatric patients needs to be examined in future studies.

## 5. Conclusions

Suicide and violent behaviors are common adverse outcomes in samples of violent offenders with schizophrenia and related disorders. This study’s findings support a lack of association between intelligence, memory, attention, and suicidality, and emphasize the importance of the executive function of planning as a risk factor for suicide attempts in violent offenders with schizophrenia spectrum disorders. This domain might play an important role in the detection of individuals who are at a high risk of suicidality. Therefore, future studies should analyze this executive domain in the clinical assessment of criminal samples of schizophrenia patients.

## Figures and Tables

**Table 1 diagnostics-10-01091-t001:** Comparison of demographic, clinical, and criminal variables between offenders with and without lifetime suicide attempts (*n* = 61).

	Suicide Attempters (*n* = 26)	Suicide Non-Attempters (*n* = 35)	*t*	*p*	Cohen’s *d*
Age	42.77 (7.57)	45.89 (8.24)	1.51	0.136	0.39
Age of illness onset					
Education (years)	9.96 (2.99)	10.89 (3.53)	1.08	0.285	0.28

	*n* (%)	*n* (%)	Chi^2^	*p*	Phi
Prior conviction	7 (26.92)	14 (40)	1.13	0.288	−0.14
Diagnosis Axis I			2.37	0.499	0.19
Schizophrenia	16 (61.53)	21 (60)			
Delusional disorder	3 (11.53)	8 (22.86)			
Schizoaffective	6 (8.57)	7 (7.69)			
Personality disorder			4.41	0.353	0.27
Antisocial	19 (54.28)	17 (65.38)			
Limit	12 (34.28)	4 (15.38)			
Other	1 (2.85)	3 (7.69)			
Family suicidality	12 (46.15)	7 (20)	4.76	0.029	0.28
Offense			3.24	0.518	0.23
Murder/Homicide	21 (80.77)	24 (68.57)			
Attempted homicide	4 (15.38)	5 (14.28)			
Sexual assaults	0 (0)	3 (8.57)			
Injuries	4 (15.38)	5 (14.28)			
Antipsychotic medication					
Conventional	6 (23.07)	12 (34.28)	0.90	0.343	−0.12
Atypical	20 (26.92)	29 (82.85)	0.33	0.564	−0.07
Both	2 (7.69)	6 (17.14)	1.17	0.280	−014
Chlorpromazine equivalents	239.8 (50–1040)	226 (100–1680)	2830.3	0.356	0.007

Phi: 0.10, 0.30, and 0.50 for small, medium, and large, respectively.

**Table 2 diagnostics-10-01091-t002:** Comparison of PANNS symptom scores and psychopathy between offenders with and without lifetime suicide attempts (*n* = 61).

	Suicide Attempters M (SD)	Suicide Non-Attempters M (SD)	*t*	*p*	Cohen’s *d*
PANNS Total	63.57 (5.39)	66.08 (6.44)	−1.65	0.104	0.42
PANNS Positive Symptoms	17.96 (2.27)	17.06 (2.31)	−1.51	0.133	0.39
PANNS Negative Symptoms	19.38 (1.83)	18.34 (1.92)	−2.13	0.037	0.55
Psychopathy Diagnosis	8.89 (4.33)	10.26 (4.42)	1.14	0.258	0.29
Psychopathy Factor 1	4.15 (2.09)	4.94 (2.87)	1.19	0.240	0.31
Psychopathy Factor 2	4.85 (2.49)	5.31 (2.55)	0.71	0.477	0.18

**Table 3 diagnostics-10-01091-t003:** Neuropsychological differences between offenders with and without lifetime suicide attempts (*n* = 61).

	Non-Suicidal (*n* = 35)	Suicidal (*n* = 26)	*t*	*p*	Adj α	Cohen’s *d*
NART Score (Estimated IQ)	103.77 (3.86)	104.69 (2.66)	−1.04	0.300	0.032	0.28
K-BIT Intelligence Total Score	90.6 (11.77)	88.65 (9.19)	0.69	0.487	0.040	0.18
K-BIT Intelligence Verbal	92.54 (11.56)	91.54 (10.1)	0.354	0.725	0.046	1.01
K-BIT Intelligence Manipulative	92.89 (12.03)	89.73 (9.98)	1.08	0.282	0.030	3.15
Verbal Fluency Test (VFT) Total	62.77 (18.43)	68.73 (15.04)	−1.35	0.183	0.022	0.35
VFT Semantic Fluency Animals	30.14 (8.35)	33.42 (6.50)	−1.66	0.102	0.014	−3.28
VFT Phonemic Fluency	33.31 (10.9)	35.31 (9.09)	−0.754	0.454	0.038	−1.99
WMS Logical Memory I	7.74 (1.65)	7.53 (1.67)	0.456	0.637	0.042	0.20
WMS Logical Memory II	6.94 (1.26)	7.00 (1.16)	−0.18	0.856	0.050	−0.05
D2 Attention (KL)	127.77 (7.70)	132.11 (3.07)	−1.235	0.222	0.024	−4.34
TMT Part A	62.97 (39.46)	48.79 (28.09)	1.56	0.124	0.018	0.41
TMT Part B	121.94 (54.64)	115.39 (65.03)	0.43	0.671	0.044	0.11
Symbol Digit Test	31.63 (7.52)	33.19 (8.02)	−0.78	0.438	0.036	0.20
Letters and Numbers Span	6.49 (1.82)	6.96 (1.34)	−1.12	0.266	0.028	0.29
Stroop Processing	39.13 (9.48)	35.78 (6.18)	1.57	0.122	0.016	0.42
Stroop Color-Word	27.93 (9.24)	26.29 (5.99)	0.79	0.433	0.034	0.21
Stroop Interference	80.49 (20.33)	70.99 (19.69)	1.83	0.073	0.012	0.47
WCST Categories Completed	4.34 (1.28)	3.92 (0.98)	1.39	0.168	0.020	0.37
WCST (Perseverative Errors)	23.54 (6.88)	24.12 (6.11)	−0.34	0.737	0.048	0.09
Zoo Map Part 1	3.74 (1.04)	5 (0.98)	−4.79	0.000	0.004	1.25
Zoo Map Part 2	5.86 (0.81)	6.42 (0.7)	−2.85	0.006	0.010	0.74
Tower of London Extra Moves	22.65 (1.49)	19.92 (2.27)	5.65	0.000	0.002	2.73
Tower of London Time Seconds	440.77 (4.88)	435.07 (7.49)	5.56	0.002	0.008	–
Hamilton Total Depression	5.11 (2.46)	7.15 (2.13)	−3.39	0.001	0.006	0.89
Hamilton Total Anxiety	6.29 (3.99)	5.23 (2.07)	1.23	0.224	0.026	0.33

Note: WMS = Wechsler Memory Test; TMT = Trail Making Test; WCST = Wisconsin Card Sorting Test. Adj α = Adjusted alpha with Benjamin–Hochberg correction.

**Table 4 diagnostics-10-01091-t004:** Results of multivariable binary logistic regression analysis (*n* = 61).

	Binary	Wald	*p*	*B*	95% CI
Lower	Upper
Family suicidality	1.01	0.75	0.386	2.74	0.28	26.62
PANNS Negative	0.06	0.04	0.853	1.07	0.55	2.08
Hamilton total depression	0.16	0.61	0.435	1.18	0.78	1.78
Zoo Map Part 1	2.05	9.29	0.002	7.75	2.08	28.89
Zoo Map Part 2	0.73	1.53	0.216	2.07	0.65	6.54
Tower of London extra moves	−0.87	8.41	0.004	0.42	0.23	0.75
Tower of London time seconds	−0.19	4.88	0.027	0.83	0.70	0.98
Nagelkerke R^2^ = 0.761

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
