# Peer review of "Neurocognitive Functioning and Suicidal Behavior in Violent Offenders with Schizophrenia Spectrum Disorders"

_diagnostics, 2020, doi:10.3390/diagnostics10121091_

Round 1
Reviewer 1 Report
The present article, titled " Neurocognitive functioning and suicidal behavior in violent offenders with schizophrenia spectrum disorders", is an interesting work that poses itself in the field of " violence and psychosis ".
Results are clearly presented.
In order to improve the manuscript, first of all I would recommend to the authors to pay more attention to language grammar and typos.
As a second step, I would suggest to revise the literature cited in the introduction and also in the discussion. I will give some suggestions of recent articles that could be cited and discussed in order to corroborate the present manuscript:
The Role of Demoralization and Hopelessness in Suicide Risk in Schizophrenia: A Review of the Literature Isabella Berardelli, Salvatore Sarubbi, Elena Rogante, Michael Hawkins, Gabriele Cocco, Denise Erbuto, David Lester, Maurizio Pompili Medicina (Kaunas) 2019 May; 55(5): 200. Published online 2019 May 23. doi: 10.3390/medicina55050200
The relationship between suicide and violence in schizophrenia: Analysis of the Clinical Antipsychotic Trials of Intervention Effectiveness (CATIE) dataset Katrina Witt, Keith Hawton, Seena Fazel Schizophr Res. 2014 Apr; 154(1-3): 61–67. doi: 10.1016/j.schres.2014.02.001
Mortality and suicide in schizophrenia: 21-year follow-up in rural China Mao-Sheng Ran, Yunyu Xiao, Seena Fazel, Yeonjin Lee, Wei Luo, Shi-Hui Hu, Xin Yang, Bo Liu, Maria Brink, Sherry Kit Wa Chan, Eric Yu-Hai Chen, Cecilia Lai-Wan Chan BJPsych Open. 2020 Nov; 6(6): e121. Published online 2020 Oct 15. doi: 10.1192/bjo.2020.106
Violent Behavior Is Associated With Emotion Salience Network Dysconnectivity in Schizophrenia Andràs Tikàsz, Stéphane Potvin, Jules R. Dugré, Cherine Fahim, Vessela Zaharieva, Olivier Lipp, Adrianna Mendrek, Alexandre Dumais Front Psychiatry. 2020; 11: 143. Published online 2020 Feb 28. doi: 10.3389/fpsyt.2020.00143
Third, please revise the sentence : "Diagnoses were confirmed by a psychologist trained in the Structural Clinical Interview for DSM-IV Axis disorders (SCID-I)". Generally, diagnoses are formulated by MDs and confirmed eventually by MDs. Probably, you were intending that a trained psychologist performed the SCID-I, which also confirmed the prior diagnosis made by a psychiatrist.
Fourth, please you are invited to always refer to the population of the study as "schizophrenia spectrum disorder patients", and not only "schizophrenia patients" since it would be incorrect. Indeed, the population under investigation includes also patients presenting delusional disorder.
Author Response
Dear reviewer,
Thank you very much for giving me the opportunity to submit a revised draft of the manuscript titled “Neurocognitive functioning and suicidal behavior in violent offenders with schizophrenia spectrum disorders to diagnosis journal. We appreciate the time and effort that you and the reviewers have dedicated to providing your valuable feedback on our manuscript. We are grateful to the reviewers for their insightful comments on our paper. We have been able to incorporate changes to reflect most of the suggestions provided by the reviewers. We have highlighted the changes within the manuscript.
Here is a point-by-point response to the reviewers’ comments and concerns.
Comment 1: In order to improve the manuscript, first of all I would recommend to the authors to pay more attention to language grammar and typos.
Response: Thank you for pointing this out. We agree with this comment. The manuscript has been checked by a native speaker of the MDPI platform. We have received the following certificate.
Comment 2: As a second step, I would suggest to revise the literature cited in the introduction and also in the discussion. I will give some suggestions of recent articles that could be cited and discussed in order to corroborate the present manuscript:
Response: We agree with this and have incorporated your articles suggestion throughout the manuscript (see in yelow).
Comment 3: Third, please revise the sentence : "Diagnoses were confirmed by a psychologist trained in the Structural Clinical Interview for DSM-IV Axis disorders (SCID-I)". Generally, diagnoses are formulated by MDs and confirmed eventually by MDs. Probably, you were intending that a trained psychologist performed the SCID-I, which also confirmed the prior diagnosis made by a psychiatrist.
Response: Thank you for this suggestion. It has been modified according to reviewer suggestion.
Comment 4: Fourth, please you are invited to always refer to the population of the study as "schizophrenia spectrum disorder patients", and not only "schizophrenia patients" since it would be incorrect. Indeed, the population under investigation includes also patients presenting delusional disorder.
Response: We have, accordingly, modified this point. We look forward to hearing from you soon regarding our submissio. Thanks for all your time and effort Sincerely
Reviewer 2 Report
This is, in summary, an interesting paper aimed to investigate the neurocognitive functioning of 61 violent offenders, most of them murders with schizophrenia and were classified as suicide attempters (n = 26) and non-attempters (n = 35). The authors found that suicide attempters demonstrated similar performance to non-attempters in a neuropsychological test across all domains of cognitive functioning, memory, attention, verbal fluency and executive functioning. They also added that, after controlling for demographic and clinical variables, suicide attempters performed better than non-attempters in two planning-related tasks: the Tower of London and the Zoo Map. Suicide attempters were characterized as having more family histories of suicidality and displaying more depressive symptoms and negative symptoms of psychopathology on the the Positive and Negative Syndrome Scale (PANSS) scale as well.
The authors may find as follows my main comments/suggestions.
First, as throughout the Introduction section, the authors correctly focused on schizophrenia, they could even stress the psychosocial disability related to invalidating disorders such as schizophrenia which are wordwide associated with stigma and discrimination. Genetic explanation of schizophrenia may potentially enhance stigma. In particular, considering schizophrenia as a genetic disorder influenced participants perception of other people's beliefs about dangerousness and unpredictability and people's desire for social distance. Importantly, a genetic explanation of schizophrenia was more frequently associated with stigmatizing attitudes. According to a study which has been published in 2013 on J Psychiatr Ment Health Nurs (PMID: 21848591), there were high levels of perceived stigmatization in medical students and medical doctors and at least half of the analyzed subjects perceived stigmatizing social attitudes against psychotic individuals. Thus, given the above information, my additional suggestion is also to rapidly include, throughout the present manuscript, the mentioned paper (PMID: 21848591). Moreover, the genetic liability of disabling conditions like schizophrenia need to be enphasized. In particular, according to a study which has been published in 2014 on World J Biol Psychiatry, gray matter reductions in the anterior cingulate have been reported as markers of genetic liability to psychosis, while reductions in the superior temporal gyrus and cerebellum may be interpreted as markers of a first onset of the illness. Thus, i suggest to briefly cite, within the main text, the specified paper on this specific topic (PMID: 22283467).
Moreover, neuropsychological assessment could be described in amore detailed manner for the general readership.
Importantly, the Discussion section has been mentioned two times within the main text. I suggest to carefully check the manuscript as first presented.
In addition, the authors could immediately present and discuss, in the first lines of the Discussion section, the most relevant study findings of this paper instead of focusing on the main aims/objectives of this study that have been adequately stressed within the main text.
Finally, what is the take-home message? While the authors reported that executive function of planning might play an important role in the detection of individuals at a high risk of suicidality, more details/information are required to this specific regard. How generally the authors hypothesize to detect patients at risk for suicide in the clinical practice using a neuropsychological perspective? How at risk subjects may be treated effectively if they exhibit neurocognitive dysfunctions?
Author Response
REVISOR 2 Thank you very much for giving me the opportunity to submit a revised draft of the manuscript titled “Neurocognitive functioning and suicidal behavior in violent offenders with schizophrenia spectrum disorders to diagnosis journal. We appreciate the time and effort that you and the reviewers have dedicated to providing your valuable feedback on our manuscript. We are grateful to the reviewers for their insightful comments on our paper. We have been able to incorporate changes to reflect most of the suggestions provided by the reviewers. We have highlighted the changes within the manuscript.
Comment 2: First, as throughout the Introduction section, the authors correctly focused on schizophrenia, they could even stress the psychosocial disability related to invalidating disorders such as schizophrenia which are wordwide associated with stigma and discrimination. Genetic explanation of schizophrenia may potentially enhance stigma. In particular, considering schizophrenia as a genetic disorder influenced participants perception of other people's beliefs about dangerousness and unpredictability and people's desire for social distance. Importantly, a genetic explanation of schizophrenia was more frequently associated with stigmatizing attitudes. According to a study which has been published in 2013 on J Psychiatr Ment Health Nurs (PMID: 21848591), there were high levels of perceived stigmatization in medical students and medical doctors and at least half of the analyzed subjects perceived stigmatizing social attitudes against psychotic individuals. Thus, given the above information, my additional suggestion is also to rapidly include, throughout the present manuscript, the mentioned paper (PMID: 21848591). Moreover, the genetic liability of disabling conditions like schizophrenia need to be enphasized. In particular, according to a study which has been published in 2014 on World J Biol Psychiatry, gray matter reductions in the anterior cingulate have been reported as markers of genetic liability to psychosis, while reductions in the superior temporal gyrus and cerebellum may be interpreted as markers of a first onset of the illness. Thus, i suggest to briefly cite, within the main text, the specified paper on this specific topic (PMID: 22283467).Moreover, neuropsychological assessment could be described in amore detailed manner for the general readership.
Response: Thank you for all your suggestion. We have now incorporated the information provided by the reviewer. We are very grateful for recommending the above papers. We have some additional ideas for future papers. thanks
Importantly, the Discussion section has been mentioned two times within the main text. I suggest to carefully check the manuscript as first presented.
Response: We have, accordingly, deleted this point.
In addition, the authors could immediately present and discuss, in the first lines of the Discussion section, the most relevant study findings of this paper instead of focusing on the main aims/objectives of this study that have been adequately stressed within the main text.
Response: We agree with this and have incorporated your articles suggestion throughout the paper.
Finally, what is the take-home message? While the authors reported that executive function of planning might play an important role in the detection of individuals at a high risk of suicidality, more details/information are required to this specific regard
Response: Thank you for this suggestion. It has been now incuded according to reviewer suggestion.
We look forward to hearing from you soon regarding our submissio. Thanks for all your time and effort Sincerely